# Ecogeographic Conditions Dramatically Affect *Trans*-Resveratrol and Other Major Phenolics’ Levels in Wine at a Semi-Arid Area

**DOI:** 10.3390/plants11050629

**Published:** 2022-02-25

**Authors:** Bat-Chen R. Lubin, Nimrod Inbar, Ania Pinkus, Maria Stanevsky, Jonathan Cohen, Oshrit Rahimi, Yaakov Anker, Oded Shoseyov, Elyashiv Drori

**Affiliations:** 1Department of Chemical Engineering, Biotechnology and Materials, Ariel University, Ariel 40700, Israel; batchenl@ariel.ac.il (B.-C.R.L.); yoni567@gmail.com (J.C.); kobia@ariel.ac.il (Y.A.); oshritra@ariel.ac.il (O.R.); 2Agriculture and Oenology Department, Eastern Regional R&D Center, Ariel 40700, Israel; anyutik81@gmail.com (A.P.); mariar@ariel.ac.il (M.S.); 3Department of Civil Engineering, Ariel University, Ariel 40700, Israel; nimrodin@ariel.ac.il; 4Department of Geophysics and Space Science, Eastern Regional R&D Centre, Ariel 40700, Israel; 5The Robert H. Smith Faculty of Agriculture, Food and Environment, The Hebrew University of Jerusalem, Jerusalem 91120, Israel; Shoseyov@agri.huji.ac.il; 6Department of Environmental Research, Eastern Regional R&D Center, Ariel University, Ariel 40700, Israel

**Keywords:** *trans*-resveratrol, wine, grapevine, phenolic compounds, terroir, climate, soil

## Abstract

Grapevines are susceptible and responsive to their surrounding environment. Factors such as climate region and terroir are known to affect polyphenolic compounds in wine and therefore, its quality. The uniqueness of the terroir in Israel is the variety of soil types and the climatic conditions, ranging from Mediterranean to arid climates. Thus, understanding the effects of climate on grapevine performance in Israel may be a test case for the effect of climate change on grapevine at other areas in the future. First, we present a preliminary survey (2012–2014) in different climate zones and terroirs, which showed that *trans*-resveratrol concentrations in Merlot and Shiraz were high, while those of Cabernet Sauvignon were significantly lower. A further comprehensive countrywide survey (2016) of Merlot wines from 62 vineyards (53 wineries) compared several phenolic compounds’ concentrations between five areas of different climate and terroir. Results show a connection between *trans*-resveratrol concentrations, variety, and terroir properties. Furthermore, we show that *trans*-resveratrol concentrations are strongly correlated to humidity levels at springtime, precipitation, and soil permeability. This work can be considered a glimpse into the possible alterations of wine composition in currently moderate-climate wine-growing areas.

## 1. Introduction

Since the early 20th century, studies have concluded that the climate is changing [1]. Historical climate records show an increase in the global mean temperature over the last 165 years, with the year 2016 reported being the hottest year on record by the World Meteorological Organization (WMO), which places the average temperature of the Earth at 1.1 °C above pre-industrial levels [2]. While these international assessments provide important insights into global processes and threats to global systems, the Levant region (Eastern Mediterranean region) is already hot, arid, and water scarce. Exhibiting extreme climatic differences in various areas, the Levant demonstrates future trends in other parts of the world in the future and may be considered a front laboratory for climate change effect [3].

The grapevine is considered one of the most sensitive cultivated plants responsive to its surrounding environment [4,5]. The growth and production of grapevines are significantly affected by environmental factors, with climate variability, primarily temperatures, affecting grapevine performance. Therefore, this species has become increasingly recognized as a bio-indicator of global warming [6].

*Terroir* are defined as highly complex ecosystems in which the grapevines (specific genomes of *Vitis vinifera*) interact with climatic, geographic, and anthropogenic conditions, affecting the grape’s chemical composition, and thus the quality and typicality of the wine produced [7]. Distinct terroir can be found even in adjacent geographic areas [8]. The terroir’ impact on the phenolic compounds in grapes, mainly consisting of flavonoids and stilbenes, has been widely investigated [9,10,11]. These compounds, as well as sugar and acid levels, which are also terroir dependent [12,13,14], play a significant role in the determination of wine color, taste and overall quality [15,16]. Therefore, it is commonly acknowledged that wine quality is terroir dependent.

Wine phenolic compounds originate mainly from the grape. The phenolic compounds are a key factor in the quality of wines in terms of color, flavor and taste [17,18]. It has also been showed that many health benefits of wine result from specific polyphenolic compounds. These compounds, such as *trans*-resveratrol, often display antioxidant activity [19], and others, such as quercetin, have been found to have numerous functions, including anti-inflammatory, antimicrobial, and anticarcinogenic properties [20,21,22].

Owing to the global importance of wine production, grapevine cultivation research led to the identification of various stilbenes, including resveratrol isomers [23,24,25]. *Trans*-resveratrol (trans-3,5,4′-trihydroxystilbene) is a natural stilbenoid compound produced by plants through the phenylpropanoid metabolic pathway [26]. It is synthesized in grapevines mainly in the grape skin [27,28], as a fundamental phytoalexin component of the biotic stress response mechanism such as fungal infection and nematode attacks, and to abiotic stress factors such as UV radiation, water stress, as an antioxidant preventing metal ion catalyzed production of reactive oxygen species [29,30,31]. *Trans*-resveratrol is suggested to be linked to many beneficial and therapeutic effects on human health [32,33,34,35], and thus, it has become an important quantitative parameter of wine’s health grade.

Israel is characterized by a dramatic climatic gradient, from the northern moderate-temperature and humidity Golan Heights to the southern hot and dry Negev. Although the entire region (Israel) is classified as subtropical with distinct seasons [36], the vineyards examined in this study represent an array of terroir (Figure 1). These vineyards represent sub-humid, semi-arid, to arid growing regions and were selected for their typical regional characteristics.

This work attempts to understand the effects of specific ecogeographic factors comprising terroir, on some wine quality parameters. Specifically, this work emphasizes the effects of specific ecogeographic factors on the levels of *trans*-resveratrol and other phenolic compounds found in commercial wines, produced from vineyards grown at distinct regions in a semi-arid area.

## 2. Results

### 2.1. Typical/Regional Trans-Resveratrol Levels in Israeli Wines

#### 2.1.1. Phase 1: Comparison between Merlot, Shiraz, and Cabernet Sauvignon (Years 2012–2014)

Preliminary characterization of *trans*-resveratrol levels in Israeli wines was performed on 130 wines from three cultivars: 34 of Merlot, 69 of Cabernet Sauvignon, and 21 of Shiraz (Figure 2A). The results show considerably high *trans*-resveratrol levels in Israeli Merlot and Shiraz compared to Cabernet sauvignon. Merlot concentrations vary between 0.42–9.28 mg/L and an average concentration of 2.52 mg/L, which is within the effective therapeutic range [37]. Shiraz wines’ *trans*-resveratrol levels concentrations vary between 0.3–6.9 mg/L, with an average concentration of 2.13 mg/L. For Cabernet Sauvignon, the average *trans*-resveratrol level is 0.56 mg/L, consisting of many samples with almost undetectable amounts.

To elucidate the possible effect of the different *terroir* found in the different regions across Israel on *trans*-resveratrol levels, the country was divided into five typical grapevine growing regions: Golan, Galilee (Galil), Central Mountain, Lowlands, and the Negev (Figure 2). Indeed, the results show a clear effect of the growing region on *trans*-resveratrol levels of Merlot wines. Significantly higher levels of *trans*-resveratrol were found in Merlot wine from the Golan (6.3 mg/L) compared to Merlots from other regions. At the Golan region, Merlot *trans*-resveratrol levels were significantly higher than those found in Cabernet Sauvignon and Shiraz wines (*p* > 0.05). Both Merlot and Shiraz wines contained higher *trans*-resveratrol levels than Cabernet Sauvignon wines at the central mountain region. In the Galil and Negev regions, the levels of *trans*-resveratrol were generally low, yet *trans*-resveratrol levels were higher in the Merlot wine compared to Shiraz and Cabernet Sauvignon (Figure 2B).

#### 2.1.2. Phase 2: Focusing on Merlot (the Year 2016)

Following these results, we widened the survey during 2016, focusing on Merlot wines from the Golan, Galil, Lowlands, Central Mountain, and the Negev regions. Eighty-three different Merlot wines were collected in Israel from 62 vineyards (53 wineries) along the above-mentioned regions. This analysis examined four additional important polyphenols found in wines; Quercetin, caffeic acid, Epicatechin, and Gallic acid. For the wines of 2016, *trans*-resveratrol levels were generally lower than in previous years. Nevertheless, we found significantly high levels of *trans*-resveratrol in Merlot wines originating in the Golan vineyards when compared to all other regions (*p* < 0.001) (Figure 3A). *Trans*-piceid, the precursor of *trans-resveratrol*, was found to be present in high levels in wines from the Golan vineyards (*p* < 0.05) (Figure 3B). Interestingly, when analyzing the levels of the other polyphenols: Quercetin, caffeic acid, Epicatechin, and Gallic acid, we did not find significant differences between the levels in wines originating from the different terroir, although significantly lower average levels of Gallic acid and quercetin were found in the southern parts compared to the Golan (*p* < 0.05) (Figure 3C–F).

Analysis of total phenolics and color intensity of the tested 2016 Merlot wines showed low levels of total phenols in the Negev wines, compared to all other regions (Figure 4A). Similarly, color intensity (CI) analysis found lower levels of color in the Merlot wine originating in the Negev (Figure 4B).

### 2.2. The Terroir Effect

To evaluate the effect of specific terroir’ conditions on changes in trans-resveratrol levels, we crosslinked various localized terroir proxies with the localized variance in the trans-resveratrol levels. The proxies that were considered are soil types, and values extrapolated from meteorological datasets for spring and summer 2016, which consisted of average daily minimal and maximal temperature and humidity, and the accumulative yearly average precipitation, calculated over all meteorological stations in each region (Table 1). We separated the growing season into spring (April–June) and summer (July–September) as the condition in Israel are very different between these two seasons. The condition at springtime generally are possible rain, higher humidity lower temp while in the summer there is no rain, low humidity and high temp.

As can be seen, the Golan Heights and Central Mountains areas are cooler, with maximal spring temperatures of 25 °C and 26 °C respectively, compared to the warmer lowlands and Negev regions, with an average maximal daily temperature of 30 °C and 32 °C, respectively. This trend continues into the summer, with maximal average daily temperature of 29 °C for the Golan Heights and Central Mountains areas, and 32 °C and 34 °C for the lowlands and Negev regions, respectively. The Negev region is also the driest region, with minimal RH in spring and summer of 20% and 30%, respectively, and the lowest accumulative rainfall. In addition, this area is more exposed than the others to solar radiation. On the other hand, the Golan Heights show the highest spring RH, and maximal rainfall.

Soil types subdivide the study area is into three regions. The dark (brown to black) basalt lithosols [38,39], are restricted to the eastern and central Golan Heights at the northernmost region. Other basaltic origin soils such as Protogrumusols and Grumusols are found mainly in the western and southern Golan Heights and several Galilee vineyards. However, most Galilee, Carmel, and Central Mountain (Judea and Samaria) vineyards are planted in Terra Rossa soil that were developed from Cretaceous limestone and dolomite and Rendzina, that was developed over Cretaceous carbonate marl and Eocene chalk outcrops and therefore defines the central region [38,39]. Finally, the arid Negev is characterized by brown litosols and light loess soils [40].

A simple classification of *trans*-resveratrol levels in vineyards, according to soil types, showed that higher *trans*-resveratrol levels were found in vineyards based on Brown basalt lithosols Soils (Figure 5). These soils are the primary soil types in the Golan Heights and were not found in other regions analyzed in this work.

Principal Component Analysis (PCA) has been used to determine the meteorological parameters with the most influence on *trans*-resveratrol concentrations (Figure 6). The PCA suggests that high relative humidity (RH) levels (89%), especially at springtime, positively affect *trans*-resveratrol levels. An additional factor found to positively affect *trans*-resveratrol levels was rain accumulation, while high spring and summer temperatures were found to have adverse effects (32 °C and 34 °C in the Lowlands and Negev respectively).

## 3. Discussion

In this work, we measured *trans*-resveratrol and other phenols levels of Merlot, Shiraz, and Cabernet Sauvignon Israeli red dry wine.

The average concentrations of *trans*-resveratrol in Israeli Merlot and Shiraz wines of 2.63 ± 0.5 mg/L and 1.94 ± 0.7 mg/L respectively, were found to be similar to those in other countries published in previous surveys (2.8 ± 2.6 mg/L and 1.8 ± 0.9 c mg/L respectively) [41,42].

In contrast, the levels of *trans*-resveratrol in Israeli Cabernet Sauvignon (0.56 mg/L ± 0.09) are significantly lower than those found in other wine countries in the mentioned research: 1.7 ± 1.7 mg/L [41]. As the health properties influence wine consumption patterns and product price [43], elevated *trans*-resveratrol levels, a potential health benefit, may influence consumers’ behavior.

The current research found a clear effect of the geographic growing area on *trans*-resveratrol levels in Merlot and Shiraz wines. The highest *trans*-resveratrol levels were found at the Golan Highest and the lowest levels at the Negev. The levels of *trans*-resveratrol in the three other areas were moderate. However, the levels of *trans*-piceid, which is the glycoside of *trans*-resveratrol [44], showed similar trends to those found for the *trans*-resveratrol levels in the wines. This may indicate that the higher or lower levels of *trans*-resveratrol in certain areas are not caused by manipulation of *trans*-piceid levels, but rather by a control mechanism located upstream to the synthesis of both. The quercetin and gallic acid levels were also found to be significantly higher at the Golan Heights than at the Negev, but the degree of change and the statistical significance were lower than those found for the *trans*-resveratrol and piceid. Other important phenolic compounds, namely caffeic acid and epicatechin, showed no regional trend and were found in similar levels at all examined regions. Analysis of the color intensity in all wine tested, and total phenolics, also showed significantly lower levels in wines produced from grapes grown at the hot Negev, which indicates lower potential wine quality in this hot region.

Given the prediction that global climate is changing, and temperatures are expected to rise [45], dramatic climate changes are possible in vast areas, including water availability, humidity, and irradiation [46,47]. Accordingly, the suitability of specific regions for quality wine production may be dramatically altered [48].

As the regions in this work represent a wide array of climatic conditions, from Mediterranean to arid, this work can promote our understanding as to possible future alterations of wine composition due to global warming, in currently moderate-climate wine-growing areas.

According to our analysis, humidity at springtime is the principal component affecting *trans*-resveratrol levels. Rain accumulation is seemingly another factor highly affecting *trans*-resveratrol levels; however, it also contributes to humidity at non-permeable soils at moderate temperatures. The connection between higher humidity and higher mildew infection on the one hand [49] and infections by mildews and elevation in *trans*-resveratrol levels as an elicitor on the other, are well established [50,51,52]. We can hypothesize that the higher levels of *trans*-resveratrol found in wines grown on highly impermeable soils, and more humid areas, are due to higher mildew infections. However, further study is needed in order to address this hypothesis.

Another factor positively affecting *trans*-resveratrol levels is rain accumulation, which is an important factor for grape wellbeing in semi-arid areas, with no summer rains [53]. Higher *trans*-resveratrol synthesis was shown to occur upon decrease in water availability to the grapevines [54], as well as at higher salinity conditions and the application of various Phytoalexin Elicitors [55].

In addition, we found that high levels of *trans*-resveratrol were found mostly in vineyards grown on the relatively heavy Brown Mediterranean Soils (basalt weathering product), which are characteristic of wide areas in the Golan Heights. Those soils are formed by weathering of basalts and tuffs and have a high clay content of heavy-textured fine Earth [56]. This can cause waterlogging after precipitation or irrigation, which may affect local conditions at the plant surrounding during critical phenological periods. Vineyards grown on other basalt origin soils, e.g., basaltic Lithosols, Protogrumusols, and Grumusols, which are far more permeable, show significantly lower *trans*-resveratrol concentrations. Similarly, samples from vineyards located on Terra Rossas and Rendzinas, found among limestone and dolomite outcrops in the Galilee, Samaria, and Judea, seldomly display significant *trans*-resveratrol concentrations. Thus, we carefully suggest that in addition to the climatic properties mentioned above, soil type and possibly its permeability act as a factor in the *trans*-resveratrol concentration measured in the final wine product, much more than other soil properties such as mineralogy which differ considerably between basalt and carbonate soil strata [28].

It should be noted that many different wineries made the wines collected, each having its own winemaking protocols. The “winery effect” sometimes resulted in varying *trans*-resveratrol levels, although the grapes were grown at very close vineyards. In future research, we aim to conduct this survey using grapes grown at the different regions but producing the wines at a research winery by a single protocol, which may focus the differences on *trans*-resveratrol levels to the terroir factor.

## 4. Materials and Methods

### 4.1. Sample Wine Collection

The survey was conducted by sampling 235 Israeli red dry wines made from Cabernet Sauvignon and Shiraz grapes (harvested and processed into wine in the years 2013–2014) and Merlot (harvested and processed into wine in the years 2013–2016) from various major grapevine growing regions across the country (Golan, Galil, Central Mountain (Judea and Samaria), Lowlands (Hashfela) and the Negev).

### 4.2. Determination of Trans-Resveratrol and Other Phenolics

In the current study, the determination of *trans*-resveratrol, *trans*-piceid, quercetin, caffeic acid, epicatechin, and Gallic acid values in wine samples was based on the methodology developed by Kerem [57], with some modifications described previously by our group [58], meant to reduce the usage of solvents, resulting in a shorter procedure and better peak detection resolution, as will be elaborated shortly. In brief, The chromatographic system consisted of a UV/Vis detector (UV-4070), RHPLC pump (PU-4180), Column oven (CO-4060), RHPLC Autosempler (AS-4150) all from Jasco, Extrema, Japan. The separation was carried out on a reversed phase column Luna 250 mm × 4.6 mm I.D. 5 u 100 A (Phenomenex). Column temperature was set at 30 °C at flow rate of 1.0 mL/min. Mobile phase consisted of 0.05% formic acid (mobile phase A) and acetonitrile (mobile phase B) using solvent gradient ranging from 5% to 35%. Concentrations were determined using calibration curves for each compound analyzed as described previously.

### 4.3. Meteorological Parameters Analysis

Meteorological data for the influencing phenological periods of vine growing, i.e., spring (April–June) and summer (July–September) 2016 was collected from weather stations located in proximity to the examined vineyards (Figure 1). Data sources are the Ministry of Agriculture and Rural Development of Israel (http://www.meteo.co.il accessed date 1 February 2021), and the Israeli Meteorological Survey (https://ims.data.gov.il/ accessed date 1 February 2021). For each vineyard location, a meteorological data set of the nearest weather-station was attributed(Figure 1): Golan Heights- Avni Eitan, Bental Loa Agar, Gamla, Marom Golan. Galilee-Baram mop, Dafna, Goren Agr, Mahanayim, Rosh Pina. Lowlands-Alonim, Bikat Hanadiv, Ein Hashofet, Newe Yaar, Revadim. Central Mountain-Ariel, Har Brach, Itamar, Rosh Tzurim, and the Negev- evivim, Sede Boker, Shani, Hazeva, Dorot, Gilat

The database includes hourly reports for several stations, and once every 10 min for other stations. The preliminary evaluation involved consistency validation and extreme data points removal.

The parameters chosen for Principal Component Analysis (PCA) are temperature, relative humidity, and precipitation. daily maxima and minima for both spring and summer seasons was collected and averaged between all relevant meteorological stations for the season (spring and summer) to generate a statistical parameter for the analysis.

Rain accumulation was calculated by summarizing the daily rainfall in each season.

### 4.4. Soils Classification

Soil classification is based on the soil map of Israel [39]. Using ArcGIS, the soil type for each vineyard has been obtained. Geographically, eastern and central Golan Heights are characterized by basaltic origin dark brown basaltic litosols Soils. Other basaltic origin soils such as Protogrumusols and Grumusols are limited to the western and southern Golan Heights and are found in several Galilee vineyards. Most of the Galilean vineyards as well as those located in the Carmel, and Central Mountain (Judea and Samaria) are planted in Terra Rossas and Rendzinas. such soils are commonly found over carbonate substratum. In the arid Negev, located in southern Israel, Loess soils are the dominant vineyard platform [40].

### 4.5. Statistical Analysis

Statistical analysis was conducted by one-way analysis of variance with Bonfferoni multiple comparison test using GraphPad Prisn version 5.02 and *p* < 0.05 was considered as significantly different.

A principal component analysis (PCA) was performed for the quantitative variables using the built-in R functions prcomp (R Core Team, 2019) and visualized using factoextra R package (Kassambara and Mundt, 2016) to create a ggplot2-based elegant visualization (Wickham, 2016).

## 5. Conclusions

To better understand the ecogeographic effects on wine quality, we studied the phenolic concentration changes in relation to the variety, the location of the vineyard, and the environmental conditions such as soil type, humidity, and temperature.

The results demonstrated that the genetic factor (variety) has preliminary effects on *trans*-resveratrol levels. We also show that the regional effect is variety-dependent and is mainly expressed in Merlot.

Phenolic analysis has shown that phenolic compound differs in their response to terroir- some such as *trans*-resveratrol, *trans* piceid were very responsive, those of quercetin and gallic acid were moderately responsive, while caffeic acid and epicatechin were not. The total phenolics and color intensity analysis indicates that Merlot wines grown at the hot Negev have lower quality than those grown at other regions.

We conclude that the main ecogeographic factors affecting trans-resveratrol levels are high relative humidity during springtime and soil type.

## Figures and Tables

**Figure 1 plants-11-00629-f001:**
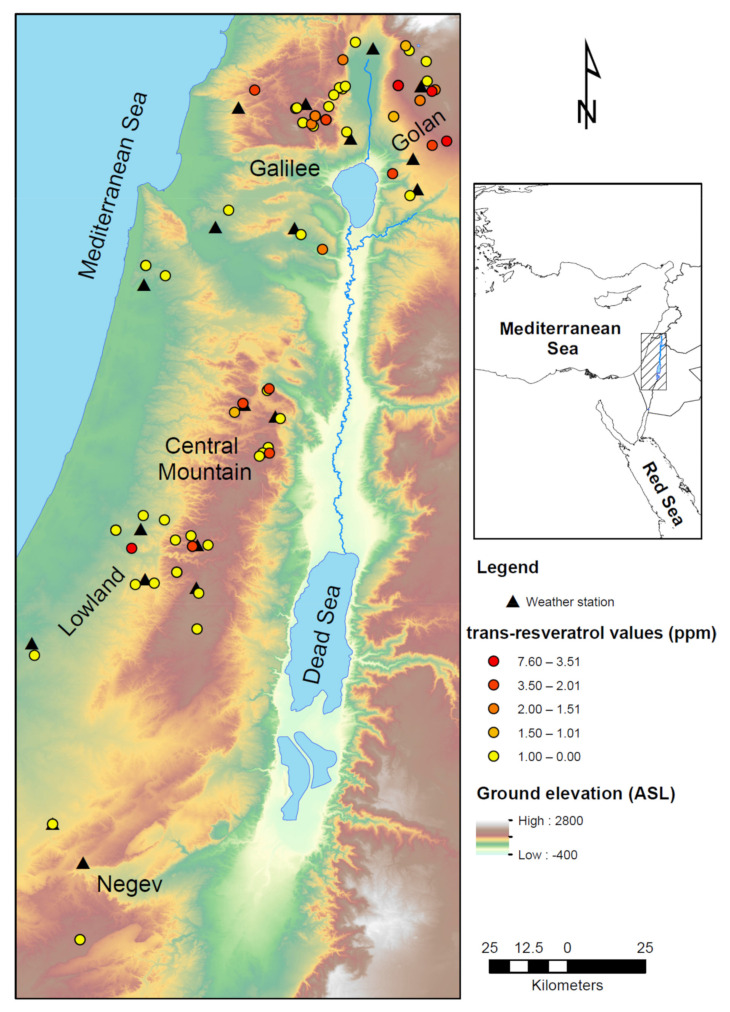
Location map of the study area. vineyards marked by circles are color coded according to *trans*-resveratrol levels (ppm). Weather stations used for climate analysis are marked with Black triangles.

**Figure 2 plants-11-00629-f002:**
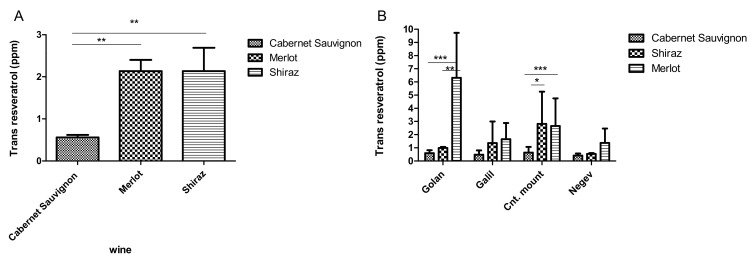
*Trans*-resveratrol level during 2012–2014 vintage (**A**). in Cabernet Sauvignon, Merlot and Shiraz Israeli wines (**B**). in wine from the different Israeli terroir, in Merlot, Cabernet Sauvignon and Shiraz varieties, * significant differences (*p* > 0.05). ** significant differences (*p* > 0.01). *** significant differences (*p* > 0.005).

**Figure 3 plants-11-00629-f003:**
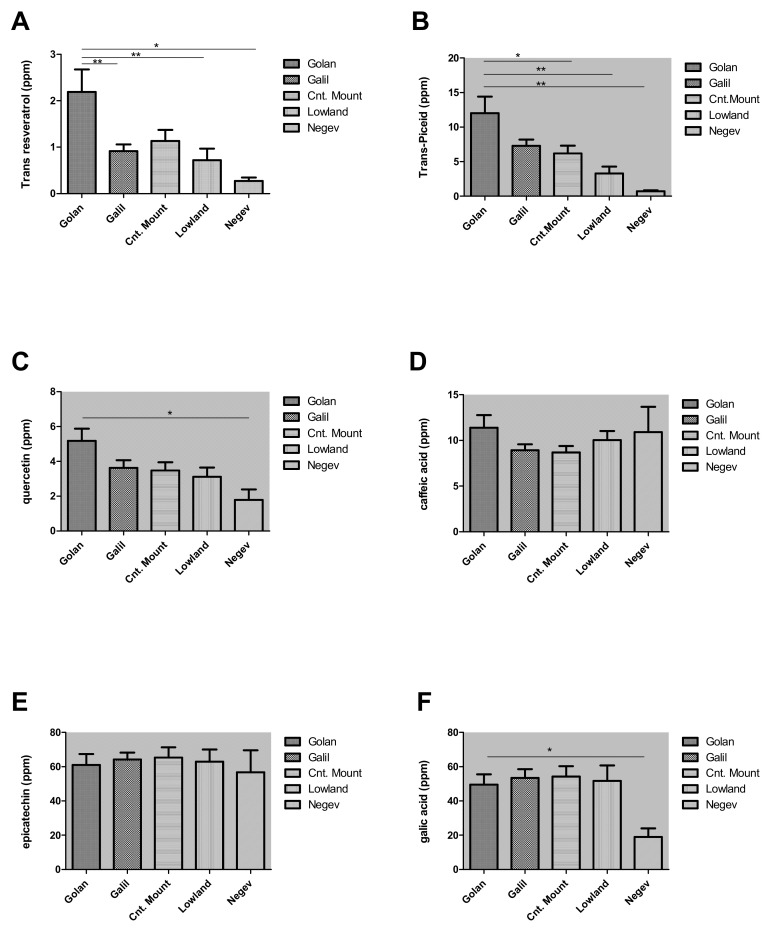
Phenolic compounds levels in Merlot wine in 2016. *Trans*-resveratrol (**A**) *trans*-Piceid (**B**), Quercetin (**C**), caffeic acid (**D**), Epicatechin (**E**) and Gallic acid (**F**). Significant high levels in *trans*-resveratrol and *trans*-Piceid were found in the Golan compared to other areas. * Significant differences (*p* > 0.05). ** significant differences (*p* > 0.01).

**Figure 4 plants-11-00629-f004:**
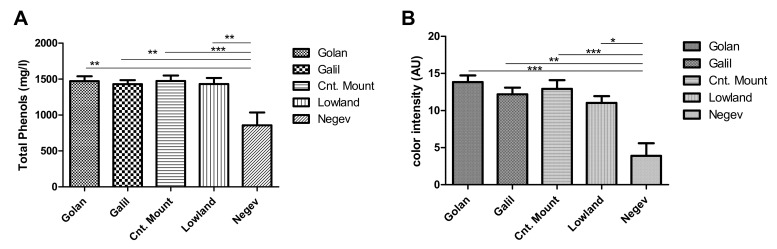
Analysis of total phenols (**A**) and Color Intensity (**B**) levels in Merlot wines originating from vineyards of the different Israeli terroir. Significant differences were found in Golan Galil and Central Mount compared to the south. * Significant differences (*p* > 0.05). ** significant differences (*p* > 0.01). *** significant differences (*p* > 0.005).

**Figure 5 plants-11-00629-f005:**
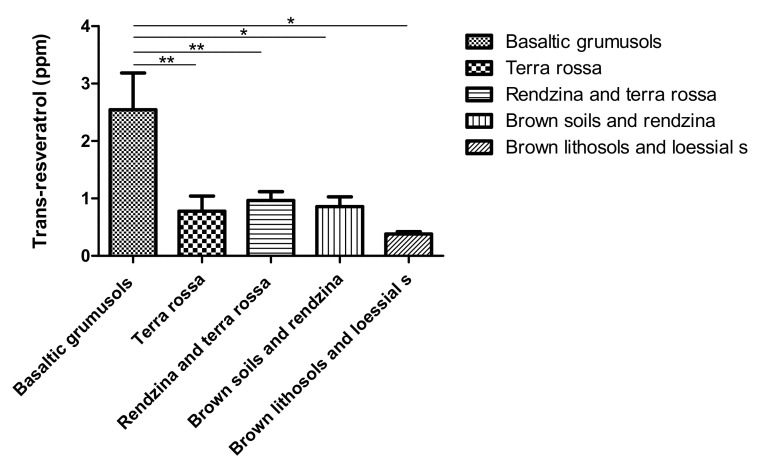
*Trans*-resveratrol concentrations according to vineyard soil type. * Significant differences (*p* > 0.05). ** significant differences (*p* > 0.01).

**Figure 6 plants-11-00629-f006:**
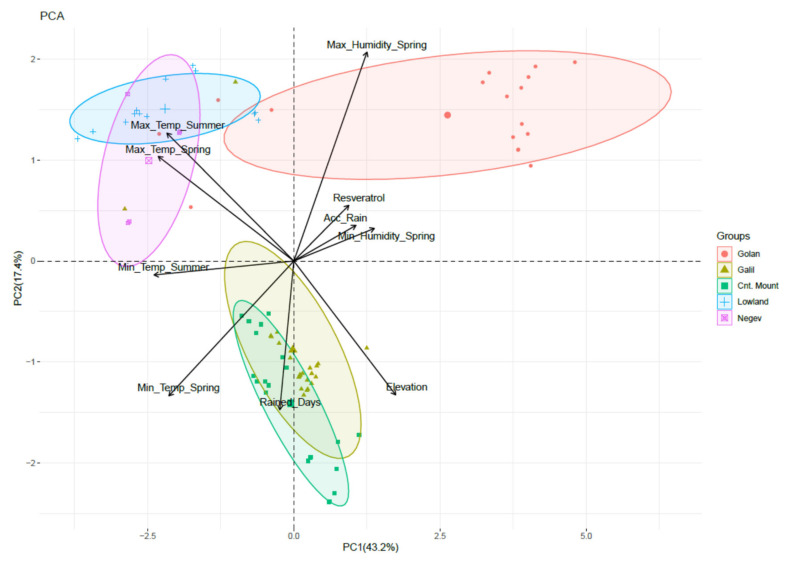
Results of Principal Component Analysis (PCA), performed on the meteorological and chemical data set. Dots represent datasets for each specific vineyard. Arrows represent the vectors deriving the separation of the dots in the PCA. the Max_Temp_Summer-Maximal temperature Summer, Max_Temp_Spring-Maximal temperature Spring, Min_Temp_Summer-Minimal temperature Summer, Min_Temp_Spring-Minimal temperature Spring, Min_Humidity_Spring-Minimal Humidity Spring, Acc_Rain-Accumulated rain.

**Table 1 plants-11-00629-t001:** Overall climatic and geographic data for the different regions.

	Galil	Golan	Lowland	Central Mountains	Negev
**Soil type**	Terra rossa, but also rendzina	Basaltic grumusols and lithosols	Brown soils and rendzina	Rendzina and terra rossa	Brown lithosols and loessial serozems
**Hydrological setting**	Moderate to strong sloping	Moderate slopes may exceed 10%.	Flat to moderate sloping	Moderate to strong sloping	mostly moderate
**Altitude (m)**	270–850	100–1200	40–320	550–930	100–800
**T _spring max._ (°C)**	27	25	30	26	32
**T _spring min._ (°C)**	15	10	15	15	15
**T _summer max._ (°C)**	30	29	32	29	34
**T _summer min._(°C)**	19	16	20	18	19
**RH _spring max._ (mm)**	73	89	82	71	80
**RH _spring min._** **(mm)**	34	34	32	27	20
**RH _summer max._ (mm)**	91	93	86	93	87
**RH _summer min._** **(mm)**	45	43	43	39	30
**Annual rainfall (mm)**	520–840	430–950	500–600	530–690	70–380
**Solar radiation (NJ/m^2^)**	18.5	19	18.7	19.7	20

## Data Availability

Not applicable.

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
