# Peer review of "Ecogeographic Conditions Dramatically Affect Trans-Resveratrol and Other Major Phenolics’ Levels in Wine at a Semi-Arid Area"

_plants, 2022, doi:10.3390/plants11050629_

Round 1
Reviewer 1 Report
REVIEW
The topic is very actual in terms of the occurrence of increasingly dry and hot periods across the world nowadays. Drought stress can influence not only primary physiological processes of plants but also the secondary metabolism. Understanding of such mechanisms may help to choose more resistant varieties of natural and also cultural plants to mitigate climate/environmental changes.
Abstract - Please overwrite the Abstract so that the same sentences are not repeated (mainly from the Discussion):
L. 202 - „The uniqueness of the terroirs in Israel is the variety of soil types and the climatic conditions, from Mediterranean to arid climates“ – the sentence is redundant here (it‘s same in the Abstract)
L. 223-225 „As the regions in this work differ in climatic conditions, from Mediterranian to arid, this work can be considered a glimpse into the possible alterations of wine composition in currently moderate-climate wine-growing areas.“ Similarly, the same was mentioned in the Abstract.
Introduction
This section is quite well written.
Please, at the end of the Introduction also formulate a hypothesis that has been confirmed by Your research/study.
Page3 – Line 57 – please specify „the Levant region“ (e.g. Eastern Mediterranean region of Western Asia…)
Results
L. 111 – please specify „the effective therapeutic range“ – (literature, citation)
Material and methods:
L. 281 - 4.1. Sample wine collection – Please, also specify the other studied phenolic compounds here.
L.294 – please specify the name of country/state (Jasco, Ex-trema – made in ???)
L.301 – 4.3 - It would be appropriate to include a table/or figure with the meteorological characteristics (annual /or for certain period average precipitation and temperatures, etc.) of the individual growing regions/study areas – for other researchers (e.g. to compare with the micro-climatic situation in South-European countries, etc.)
L.327 – Statistical analysis – PCA analysis should be described in this section
Please, specify and simplify the terms used across the text (specific ecogeographic factors comprising terroirs...Factors such as climate region and terroirs...we studied the effect of specific climatic parameters on trans-resveratrol levels...To better understand the ecogeographic effects on wine quality...The main ecogeographic factors affecting trans-resveratrol levels are humidity and soil type)
Please, correct the text technically (such are punctuation marks, big letters, missing letters etc.) - L. 286,293,299,309,310,317
Author Response
Response to Reviewer 1 Comments
Open Review
(x) I would not like to sign my review report
( ) I would like to sign my review report
English language and style
( ) Extensive editing of English language and style required
( ) Moderate English changes required
( ) English language and style are fine/minor spell check required
(x) I don't feel qualified to judge about the English language and style
|
Yes |
Can be improved |
Must be improved |
Not applicable |
|
|
Does the introduction provide sufficient background and include all relevant references? |
( ) |
(x) |
( ) |
( ) |
|
Is the research design appropriate? |
(x) |
( ) |
( ) |
( ) |
|
Are the methods adequately described? |
( ) |
(x) |
( ) |
( ) |
|
Are the results clearly presented? |
(x) |
( ) |
( ) |
( ) |
|
Are the conclusions supported by the results? |
(x) |
( ) |
( ) |
( ) |
Comments and Suggestions for Authors
REVIEW
The topic is very actual in terms of the occurrence of increasingly dry and hot periods across the world nowadays. Drought stress can influence not only primary physiological processes of plants but also the secondary metabolism. Understanding of such mechanisms may help to choose more resistant varieties of natural and also cultural plants to mitigate climate/environmental changes.
Abstract - Please overwrite the Abstract so that the same sentences are not repeated (mainly from the Discussion):
- 202 - „The uniqueness of the terroirs in Israel is the variety of soil types and the climatic conditions, from Mediterranean to arid climates“– the sentence is redundant here (it‘s same in the Abstract)
- 223-225 „As the regions in this work differ in climatic conditions, from Mediterranian to arid, this work can be considered a glimpse into the possible alterations of wine composition in currently moderate-climate wine-growing areas.“ Similarly, the same was mentioned in the Abstract.
Answer: The abstract and the sentences in the discussion were rephrased
Introduction
This section is quite well written.
Please, at the end of the Introduction also formulate a hypothesis that has been confirmed by Your research/study.
Answer: The paragraph was edited to be clearer.
Page3 – Line 57 – please specify „the Levant region“ (e.g. Eastern Mediterranean region of Western Asia…)
Answer: Eastern Mediterranean. Specify in the text
Results
- 111 – please specify „the effective therapeutic range“(literature, citation)
Answer: A reference was added
Material and methods:
- 281 - 4.1. Sample wine collection – Please, also specify the other studied phenolic compounds here.
Answer: All studied phenolic compound are specified in the next paragraph 4.2 Determination of trans-resveratrol and other phenolics
L.294 – please specify the name of country/state (Jasco, Ex-trema – made in ???)
Answer: Japan wad added
L.301 – 4.3 - It would be appropriate to include a table/or figure with the meteorological characteristics (annual /or for certain period average precipitation and temperatures, etc.) of the individual growing regions/study areas – for other researchers (e.g. to compare with the micro-climatic situation in South-European countries, etc.)
Answer: Table 1 was added to the text
L.327 – Statistical analysis – PCA analysis should be described in this section
Answer: PCA analysis moved to the statistical analysis
Please, specify and simplify the terms used across the text (specific ecogeographic factors comprising terroirs...Factors such as climate region and terroirs...we studied the effect of specific climatic parameters on trans-resveratrol levels...To better understand the ecogeographic effects on wine quality...The main ecogeographic factors affecting trans-resveratrol levels are humidity and soil type).
Answer: We disagree on this point as many ecogeographic factors were analyzed and the idea of this paper is to analyzed their specific and combined effect. Thus, we believe that generalizing terms like ecogeographic and terroir are in place.
Please, correct the text technically (such are punctuation marks, big letters, missing letters etc.) - L. 286,293,299,309,310,317
Answer: Corrected

Reviewer 2 Report
Dear Authors,
this paper is well written and brings many interesting results related to the chemical
composition of grapes in relation to ecological conditions of grape growing.
But....there is always but.....
The main objection is climate analysis. The influence of climate and soil on chemical composition of berry is discussed throughout the text, but you have not presented climatic conditions anywhere. Even soil characteristics are presented only descriptively, nowhere can we find some basic physicochemical properties of the soil.
The paper can be accepted or reconsidered after completing those requests.
Beside that here are some specific remarks:
L. 34: delete on
L.48: why only Merlot in keywords? Write the other varieties or delete all of them.
Figure 1. Resveratrol, not reservatrol
In Figure 1. we can see "Weather stations used for climate analysis" but we can not see climate analysis.
L. 266-268: Ok, but then please show some parameters like humidity, temperature and soil composition.
L.282-286: So, you can only compare ME, CS and SY from 2013-2014 vintages.
L.317: 4.4 soil classification - some facts are unnecessarily repeated.
L.328: Bonfferoni
L.337: typing mistakes
L.488-489: Journal name missing.
Author Response
Response to Reviewer 2 Comments
Review Report Form
Open Review
(x) I would not like to sign my review report
( ) I would like to sign my review report
English language and style
( ) Extensive editing of English language and style required
( ) Moderate English changes required
(x) English language and style are fine/minor spell check required
( ) I don't feel qualified to judge about the English language and style
|
Yes |
Can be improved |
Must be improved |
Not applicable |
|
|
Does the introduction provide sufficient background and include all relevant references? |
(x) |
( ) |
( ) |
( ) |
|
Is the research design appropriate? |
(x) |
( ) |
( ) |
( ) |
|
Are the methods adequately described? |
( ) |
(x) |
( ) |
( ) |
|
Are the results clearly presented? |
( ) |
(x) |
( ) |
( ) |
|
Are the conclusions supported by the results? |
(x) |
( ) |
( ) |
( ) |
Comments and Suggestions for Authors
Dear Authors,
this paper is well written and brings many interesting results related to the chemical
composition of grapes in relation to ecological conditions of grape growing.
But....there is always but.....
The main objection is climate analysis. The influence of climate and soil on chemical composition of berry is discussed throughout the text, but you have not presented climatic conditions anywhere. Even soil characteristics are presented only descriptively, nowhere can we find some basic physicochemical properties of the soil.
Answer: Table 1 was added and summarized the overall climatic and soil types, including additional verbal explanation of the climatic parameters for each region in the text (lines 176-187). " average daily minimal and maximal temperature and humidity, and the accumulative yearly average precipitation, calculated over all meteorological stations in each region (Table 1). As can be seen, the Golan Heights and Central Mountains areas are cooler, with maximal spring temperatures of 250 C and 260 C respectively, compared to the warmer lowlands and Negev regions, with an average maximal daily temperature of 300 C and 320 C, respectively. This trend continues into the summer, with maximal average daily temperature of 290 C for the Golan Heights and Central Mountains areas, and 320 C and 340 C for the lowlands and Negev regions, respectively. The Negev region is also the driest region, with minimal RH in spring and summer of 20% and 30%, respectively, and the lowest accumulative rainfall. In addition, this area is more exposed than the others to solar radiation. On the other hand, the Golan Heights show the highest spring RH, and maximal rainfall."
The paper can be accepted or reconsidered after completing those requests.
Beside that here are some specific remarks:
- 34: delete on
Answer: Deleted
L.48: why only Merlot in keywords? Write the other varieties or delete all of them.
Answer: Merlot was deleted, and two additional key words added- Grapevine, Phenolic compounds.
Figure 1. Resveratrol, not reservatrol
Answer: Figure was corrected
In Figure 1. we can see "Weather stations used for climate analysis" but we can not see climate analysis.
Answer: The climate analysis is now shown in table 1, and was shown previously in Fig 6. In the PCA analysis
- 266-268: Ok, but then please show some parameters like humidity, temperature and soil composition.
Answer: The soil types effect are shown in Fig 5 (Trans-resveratrol concentrations according to vineyard soil type) and in Fig 6. In the PCA analysis and was added and elaborated in table 1.
L.282-286: So, you can only compare ME, CS and SY from 2013-2014 vintages.
True, this is clearly stated.
L.317: 4.4 soil classification - some facts are unnecessarily repeated.
Answer: Corrected the unnecessarily sentence was deleted.
L.328: Bonfferoni
Answer: Corrected
L.337: typing mistakes
Answer: Corrected
L.488-489: Journal name missing.
Answer: journal name added

Reviewer 3 Report
Overall it is well written and very interesting with a few suggestions for improving clarity.
In general, I suggest adding a sentence of to more in the introduction about the importance and relevance of phenolic compounds in wine taste and quality (additional to the link with a human health). Also, check for consistent notation of trans-resveratrol (e.g. resveratrol, trans-resveratrol, trans-resveratrol), use global soil classification (classes) and then link to local descriptions. Use the word ‘terroir’ rather than ‘terroirs’. Check references and alignment with relevance of points being made. Please find attachment provided with further details related to suggested changes.

Author Response
Response to Reviewer 3 Comments
Review Report Form
Open Review
(x) I would not like to sign my review report
( ) I would like to sign my review report
English language and style
( ) Extensive editing of English language and style required
(x) Moderate English changes required
( ) English language and style are fine/minor spell check required
( ) I don't feel qualified to judge about the English language and style
|
Yes |
Can be improved |
Must be improved |
Not applicable |
|
|
Does the introduction provide sufficient background and include all relevant references? |
( ) |
(x) |
( ) |
( ) |
|
Is the research design appropriate? |
(x) |
( ) |
( ) |
( ) |
|
Are the methods adequately described? |
(x) |
( ) |
( ) |
( ) |
|
Are the results clearly presented? |
(x) |
( ) |
( ) |
( ) |
|
Are the conclusions supported by the results? |
(x) |
( ) |
( ) |
( ) |
Comments and Suggestions for Authors
Overall it is well written and very interesting with a few suggestions for improving clarity.
In general, I suggest adding a sentence of to more in the introduction about the importance and relevance of phenolic compounds in wine taste and quality (additional to the link with a human health). Also, check for consistent notation of trans-resveratrol (e.g. resveratrol, trans-resveratrol, trans-resveratrol), use global soil classification (classes) and then link to local descriptions. Use the word ‘terroir’ rather than ‘terroirs’. Check references and alignment with relevance of points being made. Please find attachment provided with further details related to suggested changes.
Answer: A paragraph was added to the introduction (lines 77-82) stating the relevance of phenolic compounds. Now said: " Wine phenolic compounds originate mainly from the grape. The phenolic compounds are a key factor in the quality of wines in terms of color, flavor and taste[18,19] . It has also been showed that many health benefits of wine result from specific polyphenolic compounds. These compounds, such as trans-resveratrol, often display antioxidant activity [20], and others, such as quercetin, have been found to have numerous functions, including anti-inflammatory, antimicrobial, and anticarcinogenic properties [19–21]. Terms resveratrol and terroir notation was unified.
Global soil terms were used in accordance to the global soil classification and linked to the local description.
Reviewer comments:
Overall, it is well written and very interesting with a few suggestions for improving clarity. In general, check for consistent notation of trans-resveratrol (e.g. resveratrol, trans-resveratrol, transresveratrol), use global soil classification (classes) and then link to local descriptions.
Use the word ‘terroir’ rather than ‘terroirs’. Check references and alignment with relevance of points being made.
Answer: Trans resveratrol and terroir notations were unified
Abstract Line 38-40. Include in sentence ‘.trans-resveratrol concentrations in Merlot and Shiraz were high, while those of Cabernet Sauvignon were significantly lower.’ the relationship with climate e.g. these levels were related to climate.
Answer: The sentence was added
Introduction Line 100. The legend in Figure 1 should read trans-resveratrol not resveratrol and include units on the concentrations. At the very least the figure caption should account for these omissions.
Answer: The figure legend was corrected, and the unit was added
Results Line 119. The authors need to add climate and soil type descriptions for the different wine regions identified here. I suggest a table be inserted that summarises the overall climatic and soil types (e.g. precipitation, temperature, humidity and soil classes).
Answer: Table 1 was added and summarized the overall climatic and soil types, including additional verbal explanation of the climatic parameters for each region in the text (lines 176-187). " average daily minimal and maximal temperature and humidity, and the accumulative yearly average precipitation, calculated over all meteorological stations in each region (Table 1). As can be seen, the Golan Heights and Central Mountains areas are cooler, with maximal spring temperatures of 250 C and 260 C respectively, compared to the warmer lowlands and Negev regions, with an average maximal daily temperature of 300 C and 320 C, respectively. This trend continues into the summer, with maximal average daily temperature of 290 C for the Golan Heights and Central Mountains areas, and 320 C and 340 C for the lowlands and Negev regions, respectively. The Negev region is also the driest region, with minimal RH in spring and summer of 20% and 30%, respectively, and the lowest accumulative rainfall. In addition, this area is more exposed than the others to solar radiation. On the other hand, the Golan Heights show the highest spring RH, and maximal rainfall."
Results Line 128. Figure 2b is missing an error bar in Golan Shiraz
Answer: The error bar was corrected
Results Line 159. The terroir effect (not plural) and throughout section
Answer: Corrected
Results Lines 165-172. The description of soils should make reference to a document describing soils based on a universal classification (e.g. https://www.fao.org/soils-portal/data-hub/soilclassification/fao-legend/en/). The references used do not seem to make specific reference to soils of Israel and I would strongly consider omitting these. As suggested above a table that captures all the aspects of soil and climate would provide a useful summary.
Answer: The description of soils was referenced to a document describing soils based on a universal classification https://www.fao.org/soils-portal/data-hub/soilclassification/fao-legend/en. The references were corrected and table 1 was added and text (line 176-187).
Results Line 173. Classification
Answer: Corrected to classification
Results Line 182. Please bracket what these are ‘high relative humidity levels’
Answer: We added RH to clarify, and the actual levels were added (line 219)
Results Line 184. Likewise, please bracket what these are’ while high spring and summer temperatures’
Answer: The actual levels were added (line 222)
Discussion Lines 194-196. Rephrase ‘Israeli Merlot and Shiraz average concentrations of trans resveratrol (2.63±0.5 mg/l and 1.94±0.7 mg/l respectively) were found to be similar to those in other countries published in previous surveys (2.8 ±2.6 mg/l and 1.8 ±0.9c mg/l respectively) [40,41]’ e.g. The average concentrations of trans-resveratrol in Israeli Merlot and Shiraz wines of 2.63±0.5 mg/l and 1.94±0.7 mg/l respectively, were found to be similar to those in other countries published in previous surveys (2.8 ±2.6 mg/l and 1.8 ±0.9c mg/l respectively) [40,41]
Answer: The sentence was rephrased
Discussion Line 197. Remove Nevertheless and say ‘In contrast’
Answer: Corrected
Discussion lines 199-201. Does this paper (43), highlight the relationship between trans-resveratrol concentrations, human health and wine consumption patterns and product price? If not insert correct reference or if so, place reference after this part of the sentence. 43. Bisson, L.F.; Waterhouse, A.L.; Ebeler, S.E.; Walker, M.A.; Lapsley, J.T. The Present and Future of the International Wine 446 Industry. Nature 2002, 418, 696–699, doi:10.1038/nature010
Answer: The reference highlights the importance of health beneficial of wine. moved to the right place
Discussion line 202. ‘relates to’ rather than ‘is’
Answer: Corrected
Discussion line 211. ‘found to be’
Answer: Corrected
Discussion line 217. ‘in this’
Answer: Corrected
Discussion lines 221-222. Rephrase ‘ Accordingly, the suitability of specific regions for vine growing towards quality wine production may be dramatically altered [48]’ e.g. Accordingly, the suitability of specific regions for quality wine production may be dramatically altered
Answer: The sentence was rephrased
Discussion lines 223-230. Consider removing as repetitive. Start at line 230 altered slightly, ‘According to our analyses,
Answer: Done
Discussion lines 223- 238. You make important points related to the influence of climate change on wine quality. An overarching suggestion is to keep climate observations to one paragraph and soils observations to another.
Answer: We rephrased this paragraph, and it is now clearer.
Discussion line 239. What is meant by rain accumulation? Can you briefly explain what it is?
Answer: Explanation was added in method section
Conclusions lines 269-270. ‘on trans-resveratrol levels.’
Answer: Corrected
Conclusions line 272. ‘differ’
Answer: Corrected
Conclusions lines 278-279. Rephrase ‘which levels are dramatically different at the Golan Heights than at other regions tested’. E.g. with levels that are dramatically different at the Golan Heights compared to the other regions tested’.
Answer: The sentence was rephrased
Materials and methods Check use of brackets throughout
Line 306 omit ‘Both are free online services’ highlight the importance of health beneficial of wine.
Answer: The sentence was omitted

Round 2
Reviewer 2 Report
Dear Authors,
Thank you for accepting our suggestions. The manuscript now seems appropriate fo publication in the Plants journal.
But, please reconsider (in my view) the unnecessary separation of the growing season into spring and summer (Lines: 189, 221, 350).
Also, make one sentence in Line 324.
That would be all.
Congratulations.
Author Response
Response to Reviewer 2 Comments
Dear Authors,
Thank you for accepting our suggestions. The manuscript now seems appropriate fo publication in the Plants journal.
Point 1: Please reconsider (in my view) the unnecessary separation of the growing season into spring and summer (Lines: 189, 221, 350).
Answer: We appreciate the comment though we believe that separation for seasons is important as in Israel these 2 seasons are different. While in the spring season it may still rain and the relative humidity is high. The summer season is very dry with no rain at all. Indeed, the results show the effect of high relative humidity in the spring on polyphenol levels as can be seen in fig 6. We believe that season and spring effect differently on polyphenols levels.
We added an explanation in line 179-182: We separated the growing season into spring (April- June) and summer (July-September) as the condition in Israel are very different between these two seasons. The condition at springtime generally are possible rain, higher humidity lower temp while in the summer there is no rain, low humidity and high temp.
Line 189 the spring and summer was deleted
Point 2: make one sentence in Line 324.
Answer: The sentence was rephrased. Now says: We conclude that the main ecogeographic factors affecting trans-resveratrol levels are high relative humidity during springtime and soil type.
